# TEXTURE BIAS IN PRIMATE VENTRAL VISUAL CORTEX

**Akshay V. Jagadeesh**
Department of Neurobiology
Harvard Medical School
akshay_jagadeesh@hms.harvard.edu

**Margaret S. Livingstone**
Department of Neurobiology
Harvard Medical School

## ABSTRACT

To accurately recognize objects despite variation in their appearance, humans rely on shape more than other low-level features. This is in contrast to leading deep neural network (DNN) models of visual recognition, which are texture biased, meaning they rely more on local featural information rather than global shape for categorization. Does the finding of texture bias in DNN models suggest that object representations in biological and artificial neural networks encode different types of information? Here, we addressed this question by recording neural responses from inferior temporal (IT) cortex of rhesus macaque monkeys in response to a novel object stimulus set, containing variation in shape, texture, and pose. We observed reliable tuning for both object shape and texture in IT cortex, but texture information was more accurately decodable. We assessed the performance of IT neural responses and DNN model features in classifying images based on their category or texture and found that when shape and texture were pitted against each other, IT neural responses, like DNN models, classified images based on their texture more so than their shape. However, information about both texture and shape was recoverable from the IT neural representation using a particularly tailored readout. Thus, our results suggest that the ventral visual cortex, like DNN models, provides a basis set of local visual features, and that further neural computations, perhaps downstream of IT, are necessary to account for the shape selectivity of visual perception.

## 1 INTRODUCTION

Visual object recognition requires detecting features that are diagnostic of an object's identity. For humans and other primates, the global shape of an object is perhaps the most important feature for identifying an object. Indeed, even very young children can accurately identify an object just from a line drawing of its silhouette. Extracting the global shape of an object is no trivial task, especially considering the many dimensions in which an object's local features, such as its color, pose, and surface texture, may vary. The non-triviality of extracting global shape information is best demonstrated in deep neural network models (DNNs) of visual recognition, such as Imagenet-trained deep convolutional neural networks, which are incredibly successful at the task of object categorization, but unlike humans, do so by using local visual features, such as texture, rather than global object shape (Baker et al., 2018; Geirhos et al., 2019). How does the primate visual system achieve this feat of identifying an object despite significant variation in its local visual features?

The neural system in primates most likely to underlie this ability of invariant object recognition via global shape extraction is the ventral stream of the occipitotemporal cortex, and particularly, the inferior temporal (IT) cortex. Indeed, decades of research have demonstrated that as electrical signals carrying visual information cascade along the ventral visual pathway, from primary visual cortex (V1) to IT, those neural representations become more sensitive to identity-distorting transformations and less sensitive to low-level transformations, e.g. color, texture, viewpoint, that are orthogonal to identity (Rust & DiCarlo, 2010). This hierarchical representational structure is mirrored in deep neural network models (DNNs). Like the primate ventral stream, DNNs contain a sequence of processing stages in which representations become increasingly selective for complex high-level visual features (Olah et al., 2017), and increasingly invariant to low-level variation, for example in translation, pose, or color (Goodfellow et al., 2009). In the last decade, deep hierarchical

neural network models have thus emerged as a state-of-the-art predictor of neural responses in the visual cortex of non-human primates (Yamins et al., 2014; Schrimpf et al., 2018), humans (Seibert et al., 2016; Khaligh-Razavi & Kriegeskorte, 2014), and rodents (Nayebi et al., 2023), as well as accurate predictors of object recognition behavior, at the category level (Rajalingham et al., 2018).

However, several recent findings have highlighted misalignments between deep neural network models of vision and the primate visual system (Bowers et al., 2023). For example, DNN models are highly sensitive to high spatial frequency image manipulations that are nearly imperceptible to humans (Szegedy et al., 2013). Further, Imagenet-trained DNN models fail to encode the 3D shape of objects (Jacob et al., 2021) and therefore cannot match human performance on tasks that require judging object structure from different viewpoints (Bonnen et al., 2021; O'Connell et al., 2023; Abbas & Deny, 2023; Cooper et al., 2021). Finally, Imagenet-trained DNN models rely more on texture than shape for classification (Geirhos et al., 2019; Baker et al., 2018; Hermann et al., 2020). Taken in sum, these results demonstrate that DNN models are local feature extractors, unlike primate visual perception, which critically relies on global shape information.

How then is global shape information extracted in primate vision? An emerging hypothesis suggests counterintuitively that the primate ventral visual cortex, long thought to be the seat of object perception, is instead providing a basis of local visual features that downstream neural systems can extract and recombine to generate visual object perception. Indeed, category-orthogonal information such as location and pose are more readily decodable from IT cortex than earlier regions (Hong et al., 2016); neurons in primate IT cortex are sensitive to high-spatial frequency adversarial image perturbations (Guo et al., 2022; Yuan et al., 2020); and neural populations in human ventral temporal cortex (VTC) encode the local complex visual features that make up objects, but not their spatial configuration, suggesting a texture-like representation of objects (Jagadeesh & Gardner, 2022; Ayzenberg & Behrmann, 2022). This viewpoint, if true, would suggest that DNNs are actually biologically well-aligned, not with primate visual perception as a whole, but with the initial stage of cortical processing that takes place in the ventral visual cortex.

In the present study, we offer neurophysiological evidence that indeed, representations in primate IT cortex are sensitive to local visual features corresponding to the color and surface texture of an object, more than to its global shape. We develop a novel stimulus set, designed to probe invariances to object category, identity, viewpoint, and texture, where texture is defined by spatially averaged summary statistics extracted from intermediate layers of a pretrained dCNN model. We then demonstrate that IT neurons are reliably tuned to both object category and object texture, but in most cases, texture is more readily decodable. Finally, we demonstrate that when global shape and local texture information are made to conflict in synthetic stimuli, IT neurons, like DNN models, prioritize texture over shape as a means of categorization.

## 2 METHODS

### 2.1 STIMULI

We generated a novel stimulus set that contained controlled variation in object category, exemplar, viewpoint and texture. To generate this stimulus set, we gathered 2D renderings of 3D Shapenet object models (Chang et al., 2015; Choy et al., 2016) from several different categories (e.g. car, TV monitor, sofa, bench, ship, etc.). We included multiple exemplars of objects within each category, and selected three different viewpoints for each object exemplar. We also gathered images of four different naturalistic textures — bark, brick, leaf, and elephant skin — which varied in color, pattern, and spatial frequency. We then employed neural style transfer via adaptive instance normalization to synthesize images matching the texture of each of the four naturalistic texture images and the content of each image of every object (Geirhos et al., 2019; Michaelis et al., 2019; Huang & Belongie, 2017). The set of image statistics matched by this style transfer algorithm serves as an operational definition of texture. The texture of an image is characterized as the channel-wise mean and variance (across all spatial positions) from 4 intermediate layers (relu1_1, relu2_1, relu3_1, relu4_1) of a pretrained VGG19 network. Though not explicitly included, low-level visual features such as mean color and spectral power are, to some approximation, likely encompassed within this set of texture statistics.

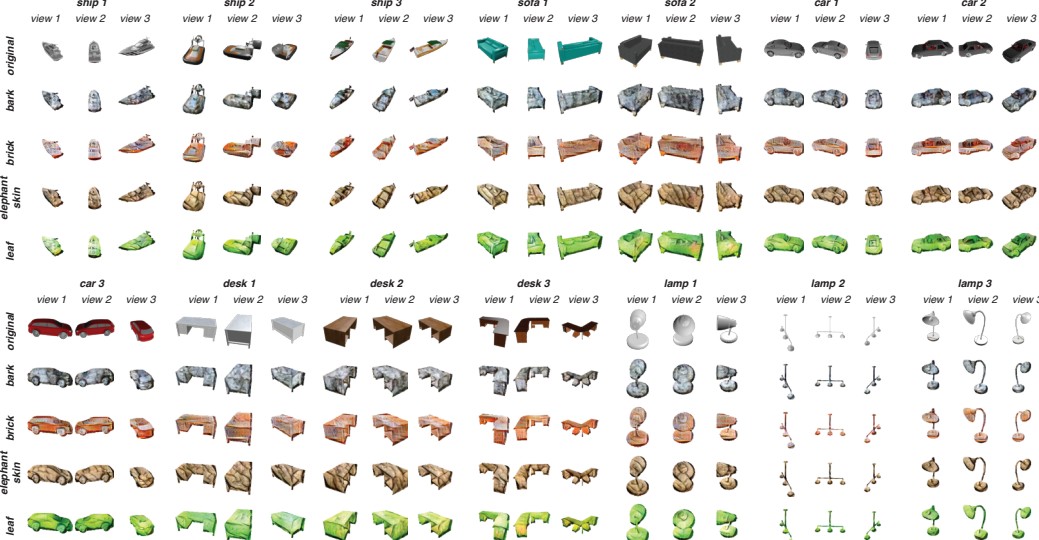

Figure 1: Stimuli used in experiment. Images are 2D renderings of Shapenet objects. Shown here is 3 different viewpoints of each object exemplar, as well as 2 to 3 different object exemplars in each category. Rows represent different textures, generated using style transfer.

## 2.2 DATA COLLECTION

Using chronically implanted electrode arrays, we recorded electrophysiological responses from the inferior temporal cortex of 6 rhesus macaque monkeys for 5 or 6 sessions per subject. 5 subjects were implanted with microwire brush arrays targeting either anterior or central IT, and one subject was implanted with a Neuropixels electrode array, targeting anterior IT. Subjects passively fixated at the center of the screen while images were presented at the center of the receptive field of the measured population. Subjects were rewarded for maintaining gaze position at the fixation point for the duration of each trial. Receptive fields of neurons in the array were mapped in a prior session. Images subtended 12 degrees and were presented for 215 ms with an interstimulus interval of 315 ms. We presented a total of 575 unique images, and each image was viewed numerous times by each subject across all sessions to allow for trial-averaging. (Repetitions per image for each monkey: Monkey R: mean=33.0, range=[31,36]. Monkey S: mean=27.1; range=[26,30]. Monkey L: mean=26.5, range=[24,28]. Monkey A: mean=15.4, range=[14,18]. Monkey B1: mean=34.0, range=[33,36]. Monkey J: mean=7.6, range=[6,10]).

## 2.3 DATA ANALYSIS

We analyzed multiunit activity, which was z-scored within each session before concatenating, to account for variability in mean and variance of response from one day to another. We calculated the split-half reliability of each unit, by randomly dividing all trials into two halves, averaging across repeats within each half, and computing the Pearson correlation between the unit's response to all 575 images in one half of the data with the other half. We selected units for inclusion in analyses only if their split half reliability exceeded 0.2.

All results shown with dynamics (Figs. 2A, 2B, 3A, 3B) were estimated by averaging over a 25 ms window, centered at 10 ms intervals between 0 and 500 ms after stimulus onset. All results plotted without dynamics (Figs. 2B, 3A, 3C-E) were estimated by summing responses over a temporal window from 100-300 ms.

All analyses were performed on an individual subject basis. For group results as we have done here, we used two methods: (1) we estimated statistics individually within each subject, then averaged across all subjects, (2) we grouped together all reliable units from all monkeys and computed each statistic on this pseudopopulation. All results shown replicated across both these methods of group-

averaging; however, for the purposes of conciseness, we have shown only results computed using the latter method.

# 3 RESULTS

## 3.1 RELIABLE, INDEPENDENT TUNING FOR TEXTURE AND CATEGORY IN IT

We found that neuronal tuning for texture was comparably reliable to tuning for category. We described neuronal tuning as the ordering of each unit's response magnitude to each category or to each texture. We computed the reliability of neuronal tuning by separately estimating the ordering of each unit's response to different categories and textures across two split-halves of the data. That is, we randomly divided all image presentations into two halves, then within each half, grouped presentations by image category, averaged together all presentations of a given category, and estimated the ordering of each unit's response magnitude over all categories (then repeated the same with image textures). We found that in both central and anterior IT cortex, on average, the reliability of texture tuning and category tuning were not meaningfully different (Fig. 2C). This is noteworthy because in our stimulus set, texture was uncorrelated with image category. Therefore, the tuning for texture cannot be explained alone by its co-occurrence with image category and suggests that there is reliable information encoded in IT cortex about low- and mid-level visual features, even when that information is unrelated to the object's identity.

## 3.2 IMAGES WITH SIMILAR FEATURES EVOKE SIMILAR NEURAL RESPONSES

The population response in IT cortex encoded information about both the texture and the shape of objects. For the purpose of dissociating texture from category, we computed the similarity of the neural response to each image with that of all other images of the same texture but different category, as well as with all other images of the same category but different texture (Fig. 3A). We also computed a baseline similarity, which was the similarity between an image and every image that did not share the same category or texture. If IT cortex were fully invariant to category-irrelevant local features, we would expect that the similarity between the neural response to images of the same texture but different categories would be indistinguishable from the similarity between the neural response to two images that differ in all aspects. Instead, we found that in both central and anterior IT cortex, images of the same texture but different category evoked a significantly more similar neural response than images that differed in their texture and category (Fig 3A,B). Moreover, in anterior IT cortex, images of the same texture but different category evoked neural response that was more similar than images of the same category but different texture, suggesting that texture information influenced the neural representation structure of aIT more than category information (Fig. 3B). In central IT cortex, images of the same category were initially represented more similarly than images of the same texture; however, by 150ms, both texture and category equally influenced the similarity structure of the cIT representation. This suggests that IT is sensitive to variation in lower-level features such as texture, even when orthogonal to category information, counter to widespread views about the invariances of category representations in IT.

## 3.3 IT ENCODES BOTH CATEGORY AND TEXTURE INFORMATION, AND WHEN IN CONFLICT, PRIORITIZES TEXTURE

To determine how distinguishable the neural representations are for different textures, different categories, or different identities, we fit linear support vector machine (SVM) classifiers on IT neural responses to predict either an object's texture, category, or object exemplar (i.e. instance-level identity) (Fig. 4A). For category, we held out all images of a particular exemplar in different views and textures, and trained the classifier on all other images of that same category and other categories, and then tested on the held out exemplar (and then repeated for all exemplars). Thus, accurate category classification required texture-invariant generalization across exemplars. For texture classification, we held out all images of a particular exemplar in a particular texture (e.g. all ship2 in leaves) then trained the classifier on the remaining images and tested on the held out exemplar x texture combination (then repeated for all exemplar x texture pairs). For exemplar classification, we held out a single viewpoint of a particular exemplar, trained the classifier on all other viewpoints of that exemplar as well as all other exemplars of that category, and tested on the heldout viewpoint

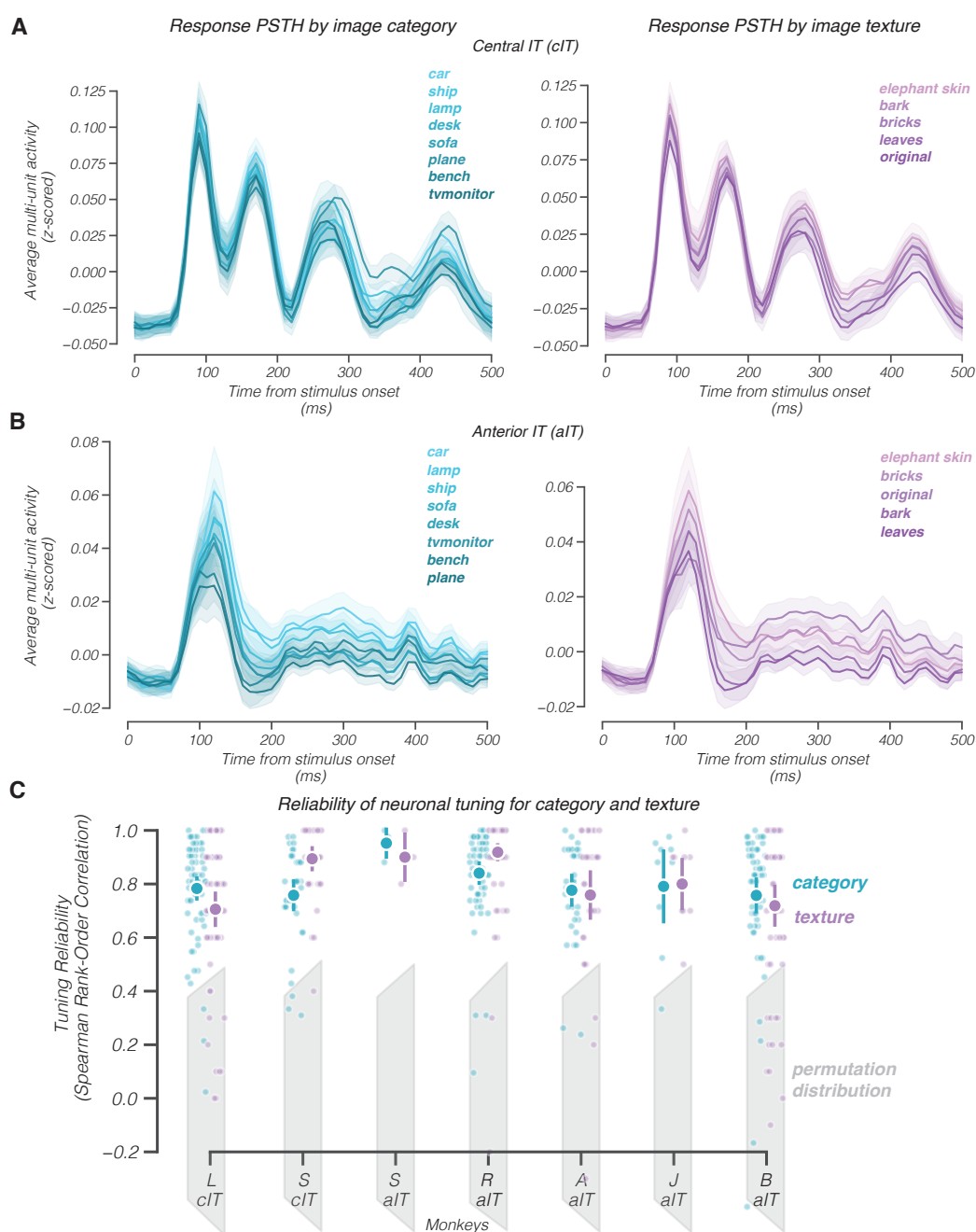

Figure 2: Tuning and preference for texture vs. category in IT cortex. (A) Post-stimulus time histograms of multi-unit activity in central IT, grouped by image category (left) and image texture (right), averaged across all reliable units from all subjects. Each time bin represents a 25ms window, plotted at 10ms intervals from 0 to 500ms after stimulus onset. (B) Same as A, for anterior IT. (C) Split-half reliability of neuronal tuning, plotted for each subject, measured by the Spearman rank-order correlation between two randomly split halves of image presentations. Each point represents one unit.

(then repeated for all viewpoints). Thus accurate exemplar categorization required discriminating between different objects of the same category. To empirically determine the chance-level performance of these classifiers, we permuted the labels and computed a null distribution of classification accuracies, a necessary step given different numbers of textures, categories, and exemplars. We also

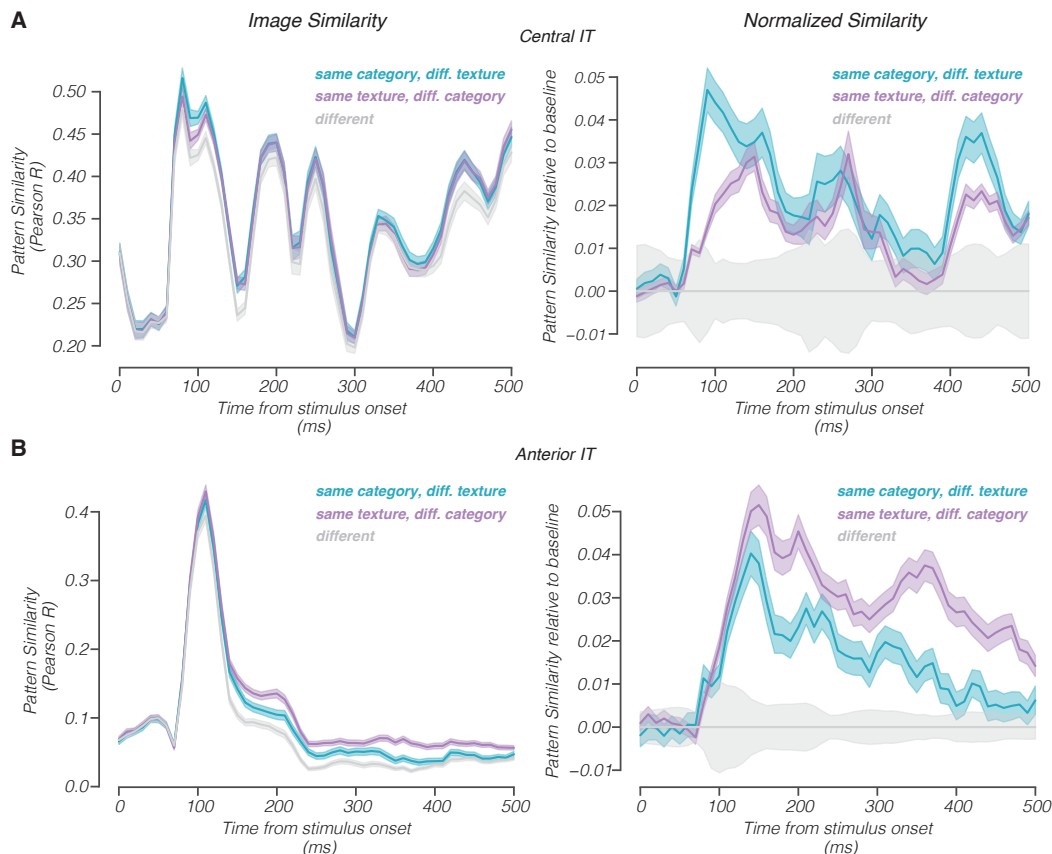

Figure 3: Preference for texture vs. category in IT cortex. (A) Similarity in neural response between pairs of images that have the same texture but different category (purple), same category but different texture (blue), and different texture and category (gray), measured by Pearson correlation. Left: raw image similarity; Right: baseline-subtracted image similarity, computed by subtracting the similarity between images with different texture and category.

performed random undersampling to assure an equal number of classes in the training set, which was necessary given that we were performing one-vs-all classification.

We found that information about object texture was more readily accessible from IT responses compared to information about object category or identity. Information about texture and category were decodable above chance in both central and anterior IT cortex (Fig 4A, blue and purple bars); however, we found that exemplar decoding was not significantly above chance-level (Fig. 4A, pink bars). Taken in conjunction with the previous finding, these results suggest that IT responses reliably encode information about low-level visual features and are at least as sensitive, if not more, to variation in low-level visual features as they are to variation in category or identity.

To compare the relative importance of texture and category information in structuring IT representations, we trained an SVM classifier to learn the optimal linear classification boundary between images of a particular category and images of a particular texture and then evaluated the classifier on images that had both that category and that texture. For example, we held out all images of ship2 in brick texture, then trained a classifier on all other ship images and all other brick images, and tested it on the held out ship2 brick images (and then repeated for all exemplar x texture pairs). We found on average that images of conflicting shape and texture were classified according to their texture label marginally more often than according to their shape label (Fig. 4B). Given that we would expect human observers to classify images according to their shape label far more often, this result demonstrates a notable divergence between IT cortex and visual perception.

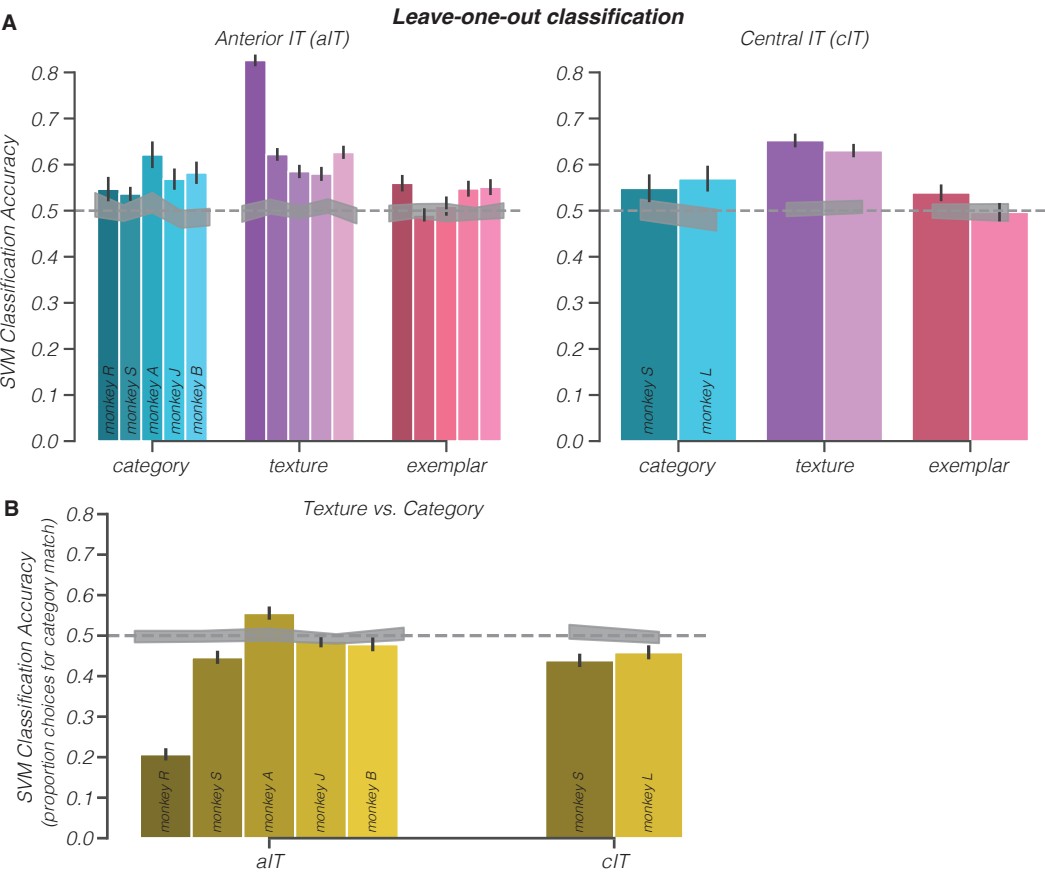

Figure 4: Classification accuracy. (A) Leave-one-out decoding accuracy of linear support vector machine classifier for category (cyan), texture (purple) and exemplar (pink) and corresponding permutation distributions (light colored filled area) in anterior IT cortex (left) and central IT (right). Each bar represents one monkey. (B) Leave-one-out classification accuracy when discriminating texture from category.

## 3.4 IT-DNN ALIGNMENT IN MATCH-TO-SAMPLE TASK

If visual object perception were supported by a direct uniform readout from IT neural responses (or DNN model features), how might the observer perform in a behavioral discrimination task? To address this question, we developed a two-alternative match-to-sample task and constructed an observer model that makes use of either neural responses or DNN features to perform the task. On each trial of this task, the observer was presented with a sample image and then given two choices. In the category matching task (Fig. 5C, left), both choices were of the same texture as the sample, but one choice matched the sample category, though was always a different object exemplar, and the other choice was an image of a different category. In the texture matching task (Fig. 5D, left), both choices were of the same category as the sample image, but one image had matching texture and the other contained a different texture. In the exemplar matching task (Fig. 5E, left), both choices were of the same category, but one image was of the same object exemplar, from a different viewpoint, and the other image was of a different object exemplar. Both choices had randomized texture. Finally, in the texture vs. category matching task, one choice ("category match") was of the same category but a different texture as the sample and the other choice ("texture match") was of the same texture but a different category as the sample. We constructed a neural observer model, which on each trial selected the choice image whose pattern of neural response was more similar (measured with Pearson correlation) to that of the sample image. Similarly, we constructed a DNN observer model, which on each trial selected the choice image whose feature vector, extracted from different intermediate layers of a DNN model, was more similar to that of the sample image.

We found that for both the category matching and texture matching tasks, the IT neural observer model was significantly above chance at selecting the correct match (Fig. 5A, blue and purple). The relatively low accuracies may be explained by the fact that each of these two tasks controlled for the effect of the opposite feature. That is, in the category matching task, both choices had the same texture, and in the texture matching task, both stimuli were of the same category. This suggests that prior results demonstrating high decoding accuracy of category from IT cortex may rely on the inherent confound in naturalistic stimuli between texture and category information. Next, for the exemplar matching task, we found that the IT neural observer model was only marginally better than chance at selecting the choice of the same exemplar in a different viewpoint (Fig. 5A, pink). Finally, for the texture vs. category matching task, we found that IT responses were significantly below chance at selecting the category match (Fig. 5a, yellow). Higher values indicate higher proportion of choices for the category match). In other words, approximately on 60% of trials, the IT neural observer model chose the image with the matching texture over the image with the matching category, providing evidence for a texture biased representation in IT cortex.

When examining performance broken down by temporal response window, we found that accuracy in the category match-to-sample task peaked around 120ms after stimulus onset (Fig. 5B, blue line), whereas texture match-to-sample task accuracy peaked later, around 180ms after stimulus onset (Fig. 5B, cyan line). In the texture vs. category matching task, the performance of the neural observer model began favoring texture after 100ms.

We found that DNN model performance was well-aligned with IT performance across the four behavioral experiments. We compared the performance of IT neurons in each of these four tasks to 7 state-of-the-art deep neural network models of vision, including 6 deep convolutional neural network models (alexnet, vgg19, inception_v3, resnet18, resnet50, and resnext50) and a vision transformer model (vit_b_16). From each model, we selected 3 intermediate layers, approximately distributed across the depth of the model (early, middle, and late). In the category matching (Fig. 5C) and texture matching tasks (Fig. 5D), we found that nearly all DNN models and layers performed above chance in selecting the correct match. Similar to the IT observer model, most DNN models performed slightly better at the texture match-to-sample task compared to the category match-to-sample task. In the exemplar matching task, most DNN models were only marginally better or no better at all than chance at selecting the matching exemplar over the image of a different object exemplar from the same category (Fig. 5E). To assess the maximum performance of DNN models with an optimal linear readout rather than an unweighted readout, we also trained SVM classifier to maximize prediction accuracy of category, texture, or exemplar information. While indeed, this did result in significantly improved category and texture discrimination for all DNN models (Fig 6A,B), exemplar discrimination was still not significantly better than chance in most DNN models (Fig. 6C). This is somewhat unsurprising considering these models are trained on the task of categorization not exemplar recognition. Nonetheless, this corroborates recent findings that Imagenet-trained DNN models lack viewpoint-invariant 3D object representations (O'Connell et al., 2023; Abbas & Deny, 2023; Jacob et al., 2021). Finally, on the texture vs. match-to-sample task, nearly all DNN models and layers were below chance in selecting the choice which matched in category (Fig. 5F) and training an SVM linear discriminator to find the optimal boundary between texture and shape did not improve models' shape bias and in many cases made DNN models appear even more texture biased (Fig. 6D). In other words, just like the IT neurons, the DNN observer models were preferentially biased towards texture-matched images over category-matched images. Interestingly, in some of the models, including alexnet, resnet18, resnet50, and resnext50, later layers of the models appeared more texture-biased than early layers.

Taken in sum, these results indicate alignment between IT neurons and DNNs in their texture-biased representation of visual objects. Indeed, across all object categories and tasks, we find a high degree of correlation in the performance of a behavioral observer using IT neural responses and one using DNN features (Fig. 4G, $R = 0.703$ for resnet50 and $R = 0.699$ for alexnet). These findings demonstrate that both IT and DNN models contain information about the shape and the texture of objects, but lack invariant representations of the 3D structure of objects. When probed with artificial stimuli that have conflicting shape and texture information, both IT neurons and DNN models preferentially represent the texture of a visual object over its shape, leading to our conclusion that IT neurons, like DNN models, are texture biased.

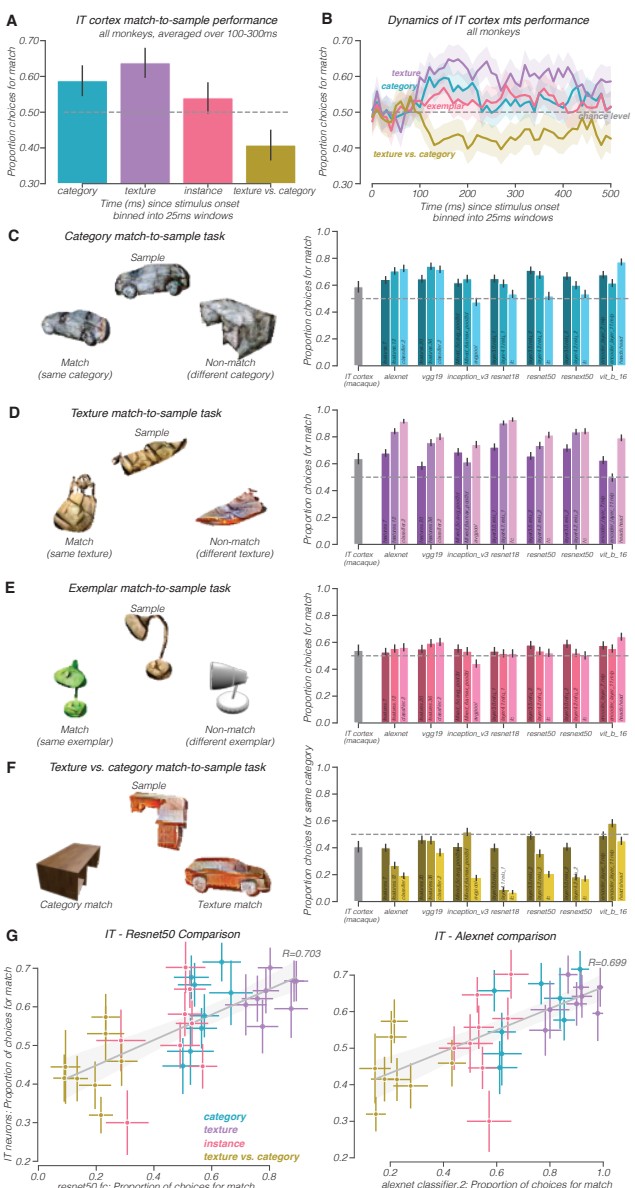

Figure 5: Match-to-sample task comparison of neural responses to DNN models. (A) Performance of IT neural observer model, constructed using neural responses pooled from 100-300 ms after stimulus onset, on each of the four tasks. Gray dotted line is chance level (50%) performance. Error bars denote standard error of the mean over trials. (B) Dynamics of IT neural observer model performance over time for each of the four tasks. (C) Category match-to-sample task. Left: example trial. Right: comparison of IT cortex performance to DNN observer model performance. Ordinate axis indicates proportion of correct choices for the matching choice. (D-E) Same as C for texture task and exemplar task. (F) Same as C for texture vs. category task, but ordinate axis indicates proportion of choices for the category match (since there is no correct choice). (G) Comparison of IT observer (abscissa) to DNN observer models (ordinate axis) (left: resnet50, right: alexnet). Each point represents one category for the sample image. Pearson correlation coefficient, R, indicated at top right of plot.

### 3.5 SHAPE-GUIDED READOUTS FROM IT CAN SUPPORT OBJECT PERCEPTION

Thus far, we have only tested unbiased readouts from IT cortex representation. However, it is possible that a linear readout that explicitly seeks a more shape-biased representation might be a closer match to perception. Indeed, we trained a logistic regression classifier explicitly designed to increase the proportion of choices for the category-matched stimulus in the 2-AFC task and tested it on a held-out subset of trials, and found that indeed this representation was significantly more shape-biased (Fig. 6E). Similarly, we trained a logistic regression classifier to explicitly increase the proportion of choices for the texture-matched stimulus and also observed a notable increase in the proportion of choices for the texture-matched stimulus. In sum, it is possible to increase shape or texture bias via a specifically tailored linear readout from the IT neural representation.

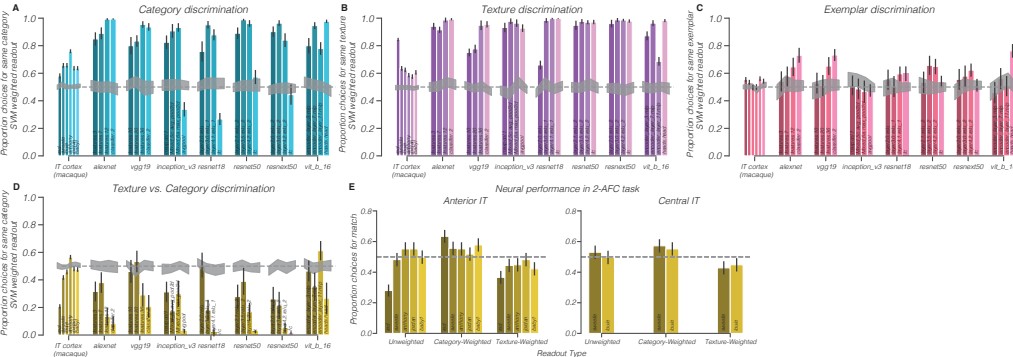

Figure 6: Readout from visual representations. (A) Optimal (SVM) weighted linear readout to maximize category discrimination from macaque brain responses and DNN features. (B,C) Same as (A) but for texture and exemplar discrimination, respectively. (D) SVM classification accuracy discriminating texture from category. (E) Comparison of unweighted readout to two types of weighted readouts - one to maximize category accuracy and one to maximize texture accuracy.

## 4 DISCUSSION

In this study, we have shown that while IT cortex reliably encodes both texture and shape information about objects, texture is preferentially represented: when texture and shape are made to conflict in artificial stimuli, IT neural responses prefer texture to shape as a basis for categorization. Further, we showed that this aligns with DNN model performance, which also encodes both texture and shape but prioritizes texture information when the two are in conflict. Finally, we provide evidence that neither IT cortex nor DNN models contain robust texture- and viewpoint-invariant 3D object representations.

Our results have two implications: first, they suggest that the primate ventral visual stream, culminating in the inferior temporal cortex, is not nearly as aligned with invariant visual object perception as has been previously assumed. This requires considering that the ventral visual stream is not the end stage of visual object processing in the brain, but simply an intermediate stage, and downstream neural circuitry is necessary for invariant object recognition. Second, these results demand a rethinking of the relationship between DNNs and visual object perception. Instead of considering DNNs as a model of perception (Bowers et al., 2023), we should consider them as a model for IT cortex, which is itself only an intermediate stage of perceptual processing.

These results provide evidence that preference for texture persists relatively late in the temporal course of the neural response. This may cohere with prior findings in the literature of non-trivial dynamics in IT responses. Sugase et al. (1999) demonstrated that early time-point responses in face-selective cells convey global, coarse, and categorical information, whereas late time-point responses convey more local, precise, and identity-based information. Our results may fit with this finding, as texture information may be more fine, local and contain high spatial frequency information, whereas category/shape information may be encoded in the global low spatial frequency component of the image.

In modeling match-to-sample task performance, we compared a uniform readout to a weighted readout from both the IT neural responses as well as from the DNN models. We found indeed that a weighted linear readout improved task performance in the category matching and texture matching, but surprisingly, not in the exemplar matching task. Moreover, an unbiased weighted linear readout did not improve shape bias in the shape-texture conflict task, but a linear readout targeted at improving shape bias was able to improve shape bias. That is, by subselecting specific units that conveyed more information about the category of the stimulus, it was possible to tailor a readout that makes the neural performance appear less texture-biased and more shape-biased. Similarly with DNNs, despite their apparent texture bias, it is possible to decode information about shape using a simple linear classifier, and simply fine-tuning the last layer can increase the shape bias of the network.

One key consideration in our study design is the importance of finding textures and shapes which evoke relatively similar strength responses in IT cortex. For example, when recording from face patches, which are highly selective for face-like stimuli, if we included faces as a shape category without finding an equivalently strong texture, our results may have instead found neurons to be shape-tuned. However, this may simply reflect the strong preference for face features, rather than actually testing claims about the preference for shape compared to texture. Indeed, recent neurophysiological evidence suggests that even in highly face-selective patches of IT cortex, neurons are largely driven by highly local features, rather than global shape structure (Waidmann et al., 2022; Sharma et al., 2023).

A limitation of our findings is that in this stimulus set, color is inherently confounded with texture. Still, the major claim of this work remains unchanged: IT neurons are not as selective for global shape information or as invariant to low- and mid-level visual features as has been previously hypothesized. Furthermore, there is evidence in the literature that the mid-level feature selectivity we observe in IT cortex is not merely about color. Long et al. (2018) showed that the functional organization of high-level visual cortex can be predicted by "texforms", grayscale stimuli that preserve local second-order statistics of natural images even while unrecognizable. Jagadeesh & Gardner (2022) controlled for low- and mid-level visual features, including color and spectral power, using a deep texture synthesis algorithm and found that neural populations in human VTC were not selective for natural feature configuration. That is, human VTC did not preferentially encode natural images compared to synthetic images with similar features in a scrambled configuration.

It is possible to achieve human-level performance in metrics of shape bias or 3D object pose invariance by training directly on the tasks which measure these abilities. For example, one can train dCNNs to be more shape-biased by incorporating stylized images (with conflicting shape and texture information) into their training diet and supervising the classification of these images according to their shape label (Geirhos et al., 2019). Similarly, it is also possible to train models to be more adversarially robust by incorporating images with adversarial perturbations into the training diet of the models (Madry et al., 2017). Finally, it is possible to train dCNN models to recognize a particular object in a novel viewpoint by exposing the model to multiple viewpoints of the same object during training and incorporating viewpoint invariance into the training objective of the model (O'Connell et al., 2023). However, our results suggest that that form of task-specific learning of object representations may not be built into the primate ventral visual cortex. Instead, the primate ventral visual cortex may provide a texture-like basis space of visual features that downstream neural circuitry uses via specifically configured readouts for specific tasks. Evidence exists for neural systems downstream of IT cortex, such as the perirhinal cortex (Bonnen et al., 2021) and the prefrontal cortex (Kar & DiCarlo, 2021) that both readout information from and provide feedback to the ventral visual cortex. Further work is required to elucidate the role that these downstream cortical regions play in closing the gap between IT cortical representations and visual perception.

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
