# OpenReview forum: "Texture bias in primate ventral visual cortex"
_ICLR.cc/2024/Workshop/Re-Align — ICLR 2024 Workshop Re-Align Poster_

### Official Review · Reviewer_LxoL · 2024-02-23
**Intruiging research question, solid methodology, and interesting results**

**Rating:** 3
**Fit:** 3
**Confidence:** 2

**Workshop Review:**

The scientific question of the work is clear and intriguing. The introduction is short, but I believe contains the necessary information for clearly motivating the research. I am not an expert in this area, therefore I can’t recommend additional studies to include in the introduction, nor can I judge the novelty of the work. However, if this test has never been done, then this is indeed an interesting piece of result that merits publication. Furthermore, the text, in general, is well-written and understandable, and the figures are nice and contain the information needed to demonstrate their results and believe their conclusions.

Pros:
- Data from 6 monkeys - it is good to see any work using electrophysiology with n>2
- Filtering units based on reliability
- Fig.2B tuning reliability clearly shows that texture is represented in IT and units are tuned to it
- Fig.3 shows that texture can be decoded to a similar extent to instance and category from the multi-unit activities
- Testing all four match-to-sample tasks for comprehensive analysis
- Comparison of IT multi-unit activities to DNNs - interesting results showing a great alignment between DNNs and IT multi-units

Cons:
- Using multi-unit activities across multiple sessions
- Repetition of images across sessions and not within sessions (if I understood correctly)
- Split-half reliability <0.2 sounds low to me but I am not an expert in this area (it would clearly be very low for behavioral data)
- Fig.3 would be great to see for separate monkeys and even for separate units similar to Fig.2B
- Match-to-sample tasks measured only Pearson correlations between responses to pairs of images. The similarity of the responses may not be equal to the similarity of representation; e.g., when encoding is not a simple linear function.
- Analyzing monkey behavior together with the neural activity could have been useful to make several point stronger. E.g., to see whether choices based on texture correlate more with IT than choices based on category.

**Reason For Not Giving Higher Score:**

All the averages across monkeys, sessions, images, and units can hinder the reliability of the results. However, I have no reason to believe that the results depended on the averaging or that these data aggregations and averaging were done incorrectly.

**Reason For Not Giving Lower Score:**

The results seem to be clear and interesting and the text is well-written.

**Reviewer Domain:**

cognitive science

---

### Official Review · Reviewer_m7t3 · 2024-02-23

**Rating:** 3
**Fit:** 3
**Confidence:** 2

**Workshop Review:**

This paper aims to clarify the features that are prominently encoded in the primate ventral stream. Prior work has shown that (1) DNNs are good models of the ventral stream (including higher areas like IT), and (2) that DNNs are biased toward encoding texture features, rather than shape/category features. This work tests the hypothesis that IT also exhibits this texture bias, despite common intuitions from the perception literature, and finds that IT does indeed exhibit this bias. Overall, this work is presented clearly, provides interesting insights into an important debate regarding the use of DNNs as models of the ventral stream, and appears fairly rigorous. Thus, I recommend that it is accepted to the Re-Align workshop.

Strengths:
See above, though I do not have the background to properly evaluate the rigor of the neural data collection and analysis. The use of a variety of neural networks models in the match-to-sample experiment is certainly another strength.

Weaknesses:
The authors note that using a uniform linear readout in the match-to-sample task is a weakness, but leave this for future work. This paper would be greatly improved by including that additional analysis in the present set of experiments. This is especially true for the DNN analyses. Rogue dimensions -- a small set of high magnitude dimensions -- are a known problem (at least for language models), and they seem likely to appear in vision models as well (https://arxiv.org/abs/2109.04404). Relevantly, these high magnitude, high variance dimensions can lead to misleading conclusions when using a uniform linear readout (such as Pearson's correlation). At least, repeating the same analysis with the spearman correlation might mitigate this potential problem.

Additionally, it would be useful to get classification accuracies for each DNN on the novel stimuli that are introduced in this work. I understand that the pretrained model's image classes may not perfectly align with the generated stimuli, but surely there is some classification accuracy that can be derived from the pretrained model. This would help contextualize the results from the match-to-sample experiment, especially when the experiment is run using activations from the later layers of the network, which are known to reflect classification information.

Typos & Style:
On Page 6: Finally, on the texture vs. match-to-sample task -> Finally, on the texture vs. category match-to-sample task

Figure 4 CDEF: Bar Graph labels extend beyond the bar

**Reason For Not Giving Higher Score:**

N/A

**Reason For Not Giving Lower Score:**

This work convincingly suggests a surprising finding - the ventral stream encodes texture features more saliently than shape features. This helps reconcile the finding that DNNs are good models of the ventral stream and the finding that DNNs have texture biases, which appear to be unlike human perceptual biases. The weaknesses that I have pointed out with the use of the DNNs can be easily rectified with simple additional analyses, rendering the paper even more convincing.

**Reviewer Domain:**

machine learning

---

### Official Review · Reviewer_P95G · 2024-02-26
**An elegant study presenting new data in favour of texture (low-level feature) bias in non-human primate IT cortex**

**Rating:** 2
**Fit:** 3
**Confidence:** 2

**Workshop Review:**

- Clarity: The paper’s aims and claims are very clear. The paper is very easy to follow and very well-written.
- Correctness: The results and analysis are correct and transparent. As acknowledged by the authors, there are some limitations to the experimental design, in particular the fact that their definition of texture includes color, which limits the scope of the interpretation (see below).
- Novelty: The paper follows a line of research focusing on texture vs. shape biases in neural networks vs. human behavior, but brings a fresh suggestion to this debate, namely that IT cortex is not as shape-biased as we might think based on behavior (i.e. is not the ‘locus’ of this behavior). The neural recordings in IT cortex with the shapenet + style transfer generated artificial stimuli in which shape and ‘texture’ (aka color) are orthogonalized are a novel contribution.
- Interest to the community: Given the general interest in texture bias as a well-known behavioral gap between DCNNs and humans, and the perhaps provocative claim that the DCNN texture bias is in fact aligned with IT, the paper will be of interest to this community.

**Reason For Not Giving Higher Score:**

- The (acknowledged) limitation here of the colour confound seems to limit the ability to attribute the representations to ‘texture’ in a narrow sense. Now it would be fair to counterargue that 'texture' is in general not well defined in this debate on texture vs. shape bias in DCNNs - I agree. But due to the choices made in the stimulus design here, color appears to be by far the most salient feature distinguishing the ‘texture’ classes, and perhaps the results thus say more about color representation than anything else, or than what we intuitively understand as texture. I also wonder if the category classification is inherently harder because of the different viewpoints. These limitations do not fully invalidate the author's claims, but do make it more difficult to pinpoint exactly what is driving the ‘texture’ decoding and corresponding neural observer behavior.
- The paper lacks information about the task given to the subjects, it would be helpful to know this to help interpret the relatively briefly lasting neural representations of category vs. texture (i.e. were any of these dimensions behaviorally relevant during recording?).
- The paper focuses exclusively on the measurement of alignment (not on bridging/improving) and the novel claims pertain to what IT is doing and may therefore mainly be of interest for neuroscientists, perhaps less for machine learning or cognitive scientists (to phrase it differently: the results of the paper do not change any views about DNNs or object recognition behaviour, only brain processing).

**Reason For Not Giving Lower Score:**

- The paper brings a fresh perspective and relevant new data to the research on gaps between DNNs and biological vision in terms of texture vs. shape bias.
- The results are very well described and well visualized and well motivated overall.
- The behavioral observer tasks are very interesting and provide a more extensive and comprehensive insight into the presence of biases in both neural responses and DNNs compared to the classic cue-conflict task from the Geirhos papers.

**Reviewer Domain:**

neuroscience

---

### Decision · Program_Chairs · 2024-03-02

Accept (Poster)